# Categorical consistency facilitates implicit learning of color-number associations

**Talia L. Retter** [ID]*, **Lucas Eraßmy, Christine Schiltz**

Department of Behavioral and Cognitive Sciences, Institute of Cognitive Science & Assessment, University of Luxembourg, Esch-sur-Alzette, Luxembourg

* talia.retter@uni.lu

## Abstract

In making sense of the environment, we implicitly learn to associate stimulus attributes that frequently occur together. Is such learning favored for categories over individual items? Here, we introduce a novel paradigm for directly comparing category- to item-level learning. In a category-level experiment, even numbers (2,4,6,8) had a high-probability of appearing in blue, and odd numbers (3,5,7,9) in yellow. Associative learning was measured by the relative performance on trials with low-probability ($p = .09$) to high-probability ($p = .91$) number colors. There was strong evidence for associative learning: low-probability performance was impaired (40ms RT increase and 8.3% accuracy decrease relative to high-probability). This was not the case in an item-level experiment with a different group of participants, in which high-probability colors were non-categorically assigned (blue: 2,3,6,7; yellow: 4,5,8,9; 9ms RT increase and 1.5% accuracy *increase*). The categorical advantage was upheld in an explicit color association report (83% accuracy vs. 43% at the item-level). These results support a conceptual view of perception and suggest empirical bases of categorical, not item-level, color labeling of learning materials.

**Data Availability Statement:** The data will be available upon publication at the following repository address: https://zenodo.org/deposit/5913254.

## Introduction

Associative learning, in which previously unconnected stimuli or events become linked, is important for behavioral responses supporting survival in a rich, dynamic environment (e.g., [1]). This fundamental type of learning is well-known for occurring even without any conscious awareness, i.e., implicitly [2–5]. While the definition of implicit is variable and debated ongoingly (e.g., [6, 7]), here we define "implicit" associative learning simply as learning that occurs incidentally, without explicit instruction or evidence of awareness during learning.

One mean through which associations can be formed is through temporal congruency, when two (or more) stimuli, features, or events occur together. For example, word stimuli appearing *consistently* in one color may become implicitly associated with that color [8, 9]. Interestingly, implicit associative learning also occurs when stimuli occur *with a higher probability* in one pairing than in any other (e.g., [10, 11]; review: [12]). This is referred to as "probabilistic learning", "contingency learning", or "statistical learning": these terms are treated equally here (but see [13–16]).

**Funding:** This work was supported by the Face Perception INTER project [INTER/FNRS/15/11015111 to CS], funded by the Luxembourgish Fund for Scientific Research (FNR, Luxembourg; http://www.fnr.lu/). The funders had no role in study design, data collection and analysis, decision to publish, or preparation of the manuscript.

**Competing interests:** The authors have declared that no competing interests exist.

The level(s) at which implicit, probability-based, associative learning occurs is actively debated. At the item-level, learning is suggested to occur for specific, individual stimuli (or small chunks of stimuli), without further abstraction (e.g., [17]; see also [18, 19]). This learning may occur either at the sensory/perceptual level, through the covariance of paired stimulus attributes that drive response predictions (e.g., [20, 21]), and/or at the response level, through the covariance of stimulus inputs and behavioral outputs (e.g., [10]; but see [22], varying the response key position).

A conceptual basis of implicit associative learning has also been proposed, in line with categorization reducing complexity in engaging with our environment (for priming: [23, 24]; sequence order regularities: [25–27]; word categories: [28]). However, evidence with probability-based learning has been limited so far: e.g., learning effects for high- vs. low-probability trials ranged from only 2-11ms and 0.7–1.8% in [28], in which participants formed implicit associations between single-exposed exemplars of word categories and their high-probability colors (color task). In [29], an effect ranging from 10-26ms and 0.5–2.9% was reported for evaluative learning of the emotional valence of a word (valence task) presented with a valence-contingent non-word, and individuals' learning effects correlated with their explicit ratings of valence. Overall, these results suggest that learning may occur at a conceptual/categorical level, although perhaps with only a small impact in addition to sensory- and response-level learning.

Here, we investigated whether categorical consistency facilitates implicit, associative learning, with a novel, direct comparison to item-level learning. In a category-level experiment, color was consistent with the numerical concept of parity: even numbers (2,4,6,8) had a high-probability of appearing in blue, and odd (3,5,7,9) in yellow. Importantly, in a control, item-level experiment, numbers were non-categorically assigned to high-probability colors (blue: 2,3,6,7; yellow: 4,5,8,9). Participants reported the parity of each number, and were not informed about color. We predicted that implicit, associative learning would occur at the category level, being evidenced by stronger learning for the parity-color consistent experiment, despite there being the same quality of sensory information (i.e., 8 digits paired with 2 colors) available across experiment levels.

We also tested whether the color associations could be explicitly retrieved, with a subsequent color association report task. The numbers were then presented in black to the participants, who were instructed to give a forced choice association with yellow or blue. This task was administered in part to rule out response learning, as participants were asked to use a different hand and with different response keys than in the main experiment, across which response-level learning is not predicted to transfer. Importantly, ability to perform this task above chance would not contradict that associative learning had occurred implicitly (e.g., [6, 30]). However, near-ceiling performance at this task could be taken as a measure of explicit awareness of color/number or color/parity relationships: therefore, this task was also administered to probe whether learning had occurred implicitly, as it should lead to above-chance but below-ceiling performance.

If implicit learning is supported by the categorical consistency of associations, as predicted, this would be in support of a conceptual nature of perception more broadly (e.g., [31–34]). More practically, it might imply that a well-known concept (color categories) might be applied to facilitate learning of a novel concept (e.g., parity categories for young children). Despite a currently lacking empirical basis of color category labeling (e.g., [35, 36]), it has been applied in a wide range of learning topics, including language (e.g., speech sounds: [37]; Reid's "color coding": [38]; language structure: [39]; see also [40]), mathematics (place values: [40]; Cuisenaire rods: [41, 42]), and text editing (syntax/semantic highlighting, e.g., in HTML, MATLAB and Python) [43, 44]. The use of color labeling for forming (implicit) associations between categories may be a particularly promising case, as opposed to coloring individual items.

## Methods

### Ethics statement

Participants gave signed, written informed consent prior to the experiment, and were fully debriefed after the experiment. The experiment was approved by the Ethical Review Panel of the University of Luxembourg (ERP 21–043), and consistent with the Code of Ethics of the World Medical Association (Declaration of Helsinki).

### Participants

Participants were 32 human adults, tested individually (age range: 18–27 years; mean age = 21.2 years; 5 males; 27 females). The sample size, divided into two groups, was determined relative to prior experiments reporting robust implicit associative learning effects (11–16 participants in [11]; 13–14 participants per group in [45]; 19–39 participants in [29]; 7 non-synesthetic participants in [46]). All reported normal or corrected-to-normal vision, and all reported to be right-handed but one (left-handed). None of the participants reported having any synesthetic experiences or learning disabilities, including dyscalculia. German was the most common first language of math education in school (N = 25), followed by French (N = 6), and Polish (N = 1). Written experiment information and instruction was in English, which was spoken by all participants, with supporting verbal explanations in English or German, as requested. Participants were recruited from the University of Luxembourg community through educational and social media, for a study on numerical cognition (with no mention of color or parity).

### Stimuli and materials

The stimuli were eight Arabic numerals: 2, 3, 4, 5, 6, 7, 8, and 9. They were presented in Arial font, at an average height of 3.1˚ of visual angle. The size varied across stimulus presentations, in 7 steps spanning ±20% of the average size, in order to prevent low-level, pixel-based habituation. The blue and yellow font colors were presented in their maximal red, blue, green (RGB) monitor subpixel representations (blue = 255/255 B; yellow = 255/255 R + 255/255 G).

The experiment was built and executed in PsychoPy3 v2020.2.8 [47], running over Python (Python Software Foundation, USA). Stimuli were presented on a Dell S2419HGF (USA) monitor, with a screen size of about 53x30 cm and a refresh rate of 120 Hz, connected to a Dell (USA) PC fitted with a GeForce 1050 (Nvidia, USA) graphics card. Data were analyzed with SPSS Statistics 27 (IBM, USA).

### Procedure

Participants performed a parity task in the main experiment, i.e., they were instructed to report as accurately and as fast as possible whether a number presented on the screen is odd (up arrow) or even (down arrow), with the middle and index fingers of their dominant hand, respectively. Number stimuli were presented in either blue or yellow: to enable implicit learning, each stimulus was presented more often in one color (at an average ratio of 10:1). Trials in which the number is presented in its high-probability color are termed *congruent* trials (vs. *incongruent* trials with the low-probability color). Importantly, participants were neither informed about nor required to respond to stimulus color (e.g., [11, 28, 29, 45, 46]).

There were two levels tested: *category-level* and *item-level*, defining two versions of the main experiment (**Fig 1A**). These experiment levels were tested between participants (16 participants in each group, through random assignment). In the category-level experiment, high-probability digit colors were parity-consistent, i.e., the digits 2,4,6, and 8 were presented with a

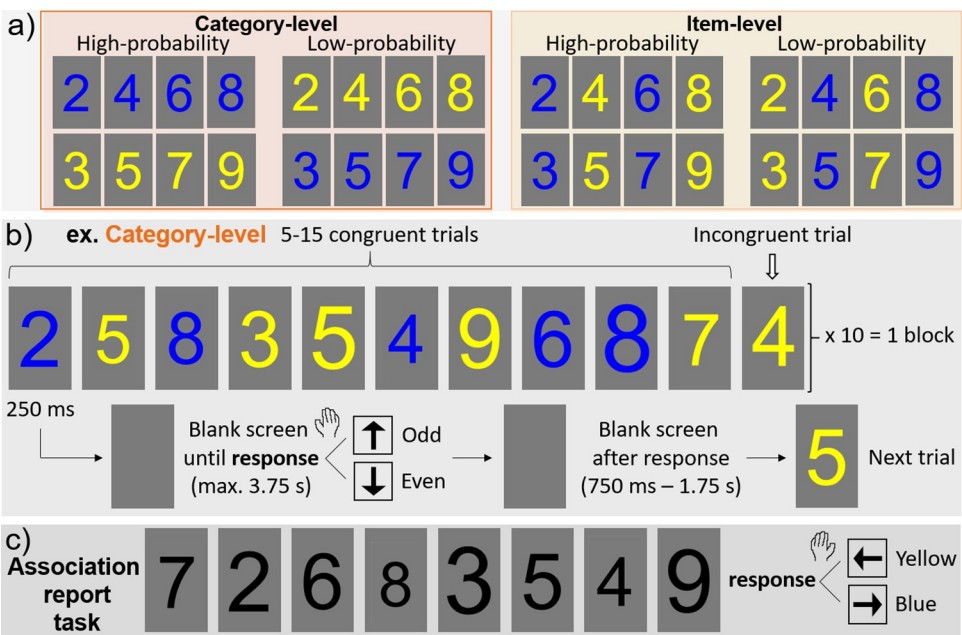

**Fig 1. Experimental design. a)** There were two levels of the main experiment: category-level (parity-consistent) and item-level, defined by their color-number pairings in high-probability and low-probability appearances. **b)** The stimulation paradigm, for both levels (congruent trials = high-probability color-number pairings; incongruent trials = low-probability color-number pairings). **c)** Following the main experiment, participants completed a color association report task (with trial timing as in panel b).

high probability (p = .91) in blue, and 3,5,7, and 9 with a high probability in yellow (the opposing color assignment was low-probability, p = .09). In the item-level experiment, the digits 2,3,6 and 7 were presented with high-probability in blue, while 4,5,8 and 9 with high-probability in yellow (again, the opposing color assignment was low-probability).

In each trial, a number was presented in the center of a grey screen (128/255 RGB) for 250ms, after which the blank grey screen persisted (**Fig 1B**). If no response was given, the next trial would begin after 3.75s, for a maximum SOA of 4s. Following a response, there was a wait period of a blank gray screen, jittered between 750ms-1.75s, before the next trial. In each block, there were 5–15 congruent (i.e., high-probability color-number pairing) trials presented in between each incongruent (i.e., a low-probability color-number pairing) trial, for a total of 10 incongruent trials. Each block lasted about three minutes, and there were five blocks in total in the experiment.

After the main experiment, there was an association report task (e.g., [46]), in which participants were asked to explicitly retrieve the color associated with each of the number stimuli (**Fig 1C**). Participants were instructed to report whether the number, presented in black (0/255 RGB) font, was associated more with yellow (left arrow) or blue (right arrow), by responding as accurately and as fast as possible with the index and middle fingers of their non-dominant hand (i.e., the opposite hand as used in the main experiment). Numbers were otherwise presented with the same trial design as in the main experiment. The eight numbers were presented in a random order, and this randomization was repeated two additional times, for a total of 24 trials, presented in a single block.

Finally, to assess mathematical fluency, participants completed the paper-and-pencil Tempo-Test Rekenen (TTR; [48]; as in [49]). This test is comprised of five subtests, each consisting of 40 mathematical problems (1: addition; 2: subtraction; 3: multiplication; 4) division; 5) mixed). Participants were given a maximum of 1 minute per subtest to solve as many

problems as possible, in the order given. The score on the test is equal to the sum of all correctly solved problems (for a maximum of 200). Note that individuals' TTR scores did not significantly differ across the randomly assigned experimental groups, $t_{30} = 0.68$, $p = .25$, $d = 0.24$ (category-level: $M = 144.5$; $SD = 24.28$; item-level: $M = 138.9$, $SD = 22.60$).

### Data analysis

In order to measure the potential learning of color-number associations, a potential Stroop-like interference effect ([50]; see Discussion) is measured in terms of decreased performance (i.e., increased response time (RT) and/or decreased accuracy) for incongruent relative to congruent trials. For response time (RT) of correct trials only, scores were excluded that were over 2.5 SDs from each individual's mean (excluding 2.3% of trials on average). Data were analyzed separately for accuracy and RT, with a repeated-measures analysis-of-variance (ANOVA) with a within-participants factor of *congruency* (congruent and incongruent trials), and a between-participants factor of experiment *level* (category-level and item-level). Following significant interactions, follow-up *t*-tests were conducted within each experiment level, to test the prediction that performance was better for congruent than incongruent trials, with one-tailed, paired samples *t*-tests. For the color association report task data, one-tailed, independent samples *t*-tests were applied to compare experiment levels; one-tailed, one-sample *t*-tests were applied to test whether each experiment sample was above 50% chance-level. *T*-tests were also applied on congruency difference scores, i.e., incongruent–congruent accuracy, and congruent–incongruent RT. Numerical fluency (TTR) scores were correlated with individuals' difference scores; for RT, one individual was removed from the analysis for a difference RT score greater than 2.5 SDs from the sample mean. In the case of a violation of Levene's Test for Equality of Variances, unequal variances *t*-tests were used with corrected degrees of freedom.

### Results

There was an interaction between *congruency* and experiment *level*, for both accuracy, $F_{1,30} = 16.4$, $p < .001$, $\eta_p^2 = 0.35$, and response time (RT), $F_{1,30} = 4.59$, $p = .040$, $\eta_p^2 = 0.13$. For the category-level experiment, in line with a predicted decrease in performance for incongruent trials, these responses were 8.3% less accurate, and 40ms slower, than those to congruent trials (**Fig 2A**). These differences were both significant: accuracy, $t_{15} = 3.58$, $p = .001$, $d = 0.89$; and RT, $t_{15} = 2.89$, $p = .006$, $d = 0.72$. In comparison, for the item-level experiment (**Fig 2B**), there was a small delay for low-probability trials (8.7ms mean difference), $t_{15} = 1.80$, $p = .046$, $d = 0.45$, countered by a 1.5% *more* accurate response for low-probability than high-probability trials, $t_{15} = -2.29$, $p = .018$, $d = -0.57$.

Across experiment versions, the congruency difference scores were significantly greater for the category-level than item-level experiment, for both accuracy (congruent–incongruent), $t_{17} = 4.05$, $p < .001$, $d = 1.43$, and RT (incongruent–congruent), $t_{30} = 2.14$, $p = .020$, $d = 0.76$ (**Fig 3**). Indeed, for the category-level experiment, 11/16 participants had difference scores in the expected direction for *both* accuracy and RT (i.e., were in quadrant I of Fig 3); but this applied to only 2/16 participants in the item-level experiment. Most participants (9/16) in the item-level experiment had positive RT difference scores at the expense of accuracy (i.e., were in quadrant II of Fig 3).

The results were examined across the main experimental blocks (**Fig 4**). For the category-level experiment, in terms of accuracy, the main effect of *congruency* was confirmed, $F_1 = 12.9$, $p = .003$, $\eta_p^2 = 0.46$, with incongruent accuracy being lower overall. However, there was no main effect of *block*, $F_4 = 0.77$, $p = .55$, $\eta_p^2 = .049$, and no interaction between these factors, $F_{1,4} = 1.27$, $p = .29$, $\eta_p^2 = 0.078$. A planned comparison of congruent and incongruent trials in the first block suggests that the associations were not present at experiment onset, $t_{15} = 1.09$, $p =$

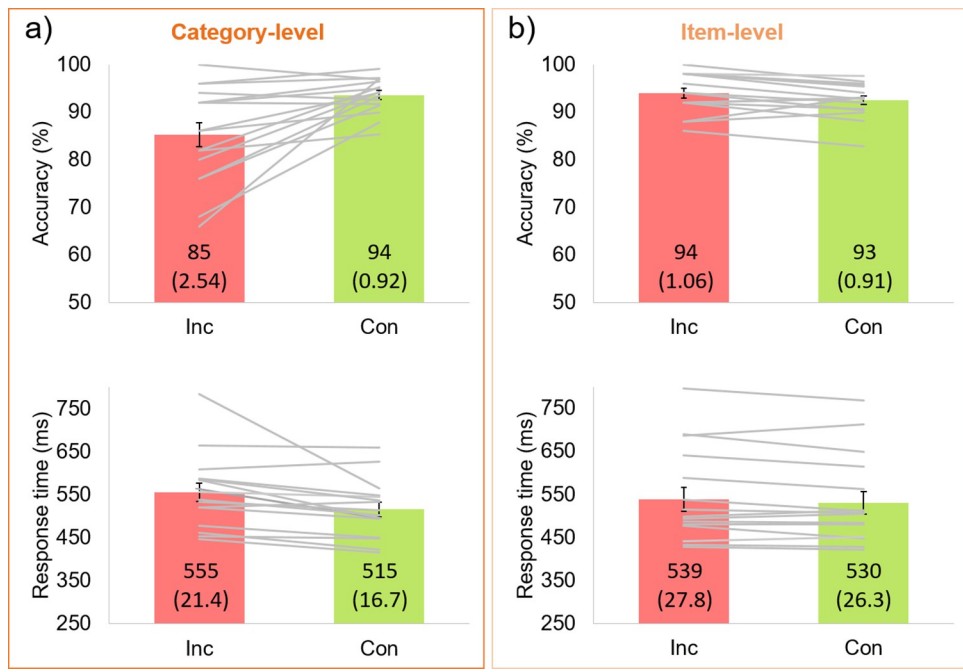

**Fig 2. Results in terms of accuracy (%) and response time (ms).** Individual participants are represented as lines above the group-level mean bar plots, with error bars of ±1 SE. Exact mean and SE values (in parentheses) are given on each bar. **a)** The category-level experiment, for which incongruent (Inc) trial responses were 8.3% less accurate and 40ms slower than congruent (Con) trials. **b)** The item-level experiment, for which incongruent trial responses were 1.5% more accurate and only 8.7ms slower than those to congruent trials.

.15, $d = 0.27$. For RT, the main effect of *congruency* was also confirmed, $F_1 = 8.75$, $p = .010$, $\eta_p^2 = 0.37$, and while there was no main effect of *block*, $F_4 = 0.46$, $p = .77$, $\eta_p^2 = 0.03$, there was additionally a significant interaction, $F_{1,4} = 3.50$, $p = .012$, $\eta_p^2 = 0.19$, between these two factors. Follow-up comparisons of congruent and incongruent trials showed no differences in the first two blocks ($t$'s $<1.6$, $p$'s $> .068$, $d$'s $< 0.40$), but significant differences in the last three blocks ($t$'s $> 1.86$, $p$'s $< .042$, $d$'s $> 0.46$).

## Association report task

Following the main experiment, participants reported the high-probability (i.e., congruent) color with 83% accuracy (62–100% range across participants) in the category-level experiment, but with only 43% accuracy (8–92% range) following the item-level experiment, $t_{22} = 5.65$, $p < .001$, $d = 2.0$ (**Fig 5A**). This accuracy was significantly above chance level in the category-level experiment, $t_{15} = 10.2$, $p < .001$, $d = 2.6$, but not the item-level experiment, $t_{15} = -1.2$, $p = .13$, $d = -0.29$. Participants were also faster to respond following the category-level than item-level experiment, 859 *vs.* 984ms, although this difference was not significant, $t_{30} = 1.16$, $p = .13$, $d = 0.41$. An analysis of individual stimulus items showed that there was only one participant, in the category-level experiment, with perfect accuracy for all eight items (**Fig 5B**). The average number of items with perfect accuracy was 4.7 (SE = 0.57) in the category-level *vs.* 1.1 (SE = 0.39) in the item-level experiment.

## Correlations

Association report accuracy positively, significantly correlated with individuals' congruency difference scores in terms of both accuracy (congruent–incongruent), $r_{14} = .63$, $p = .004$, and

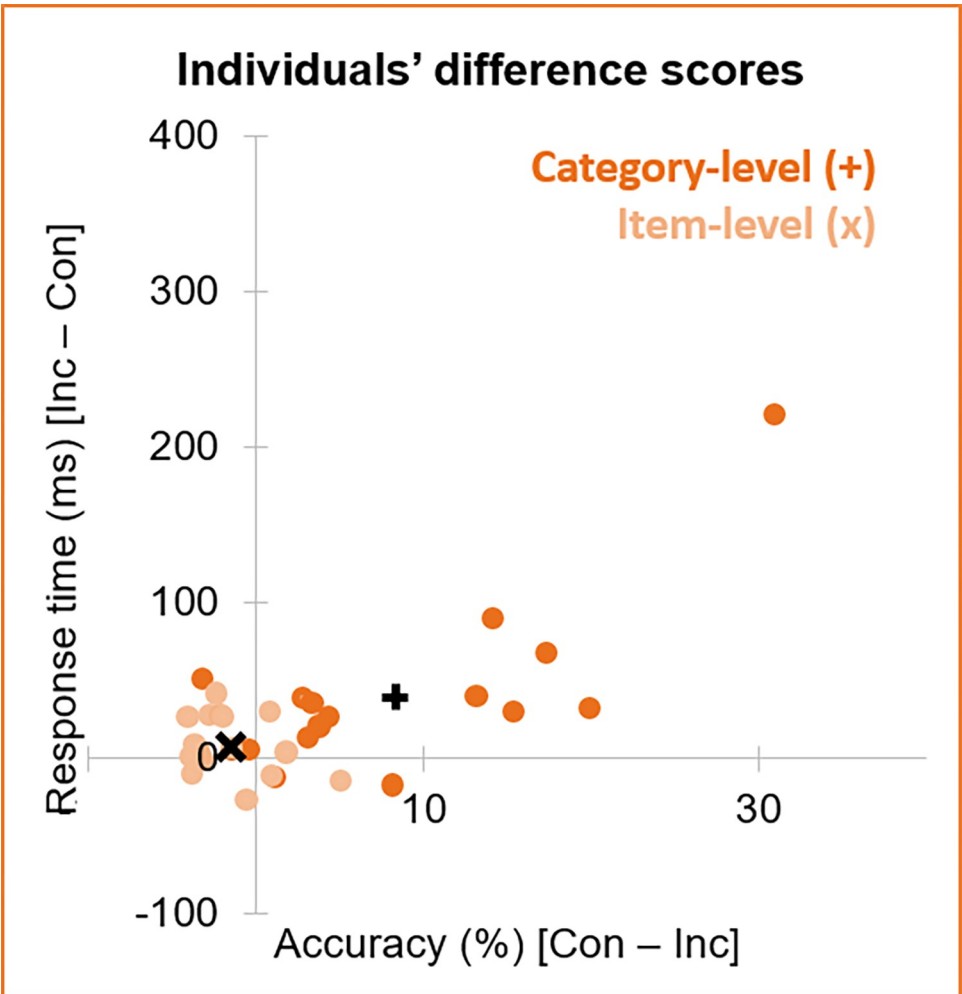

**Fig 3. Individuals' differences scores.** Difference scores (dots) are given for each individual in terms of accuracy (%) and response time (ms) for the category-level (dark orange) and item-level (peach) experiment versions. The mean difference score for the category-level experiment (black +) is 8.3% by 40ms; for the item-level experiment (black x), it is -1.5% by 8.7ms. Key) Inc = Incongruent; Con = Congruent. Quadrants are numbered counter clockwise from I in the upper right.

RT (incongruent–congruent), $r_{13} = .54$; $p = .019$, in the category-level experiment (**S1 Fig**). These correlations were neither present for the item-level experiment, nor for the association report RT of both experiment versions ($p$'s > .35). Regarding individuals' numerical fluency (TTR) scores, there were positive, although non-significant, correlations with individuals' congruency difference scores in the category-level experiment: accuracy, $r_{14} = .18$, $p = .26$, and RT, which neared significance, $r_{13} = .42$, $p = .062$.

## Discussion

### Category-level associative learning

Consistent parity-color associations enabled rapid, implicit learning of color-number pairings. Category-level experiment performance was reduced for incongruent vs. congruent trials: 8.3% less accurate and 40ms slower, and most participants (~70%) showed an effect in both variables. These findings confirm that categorical consistency facilitates implicit associative

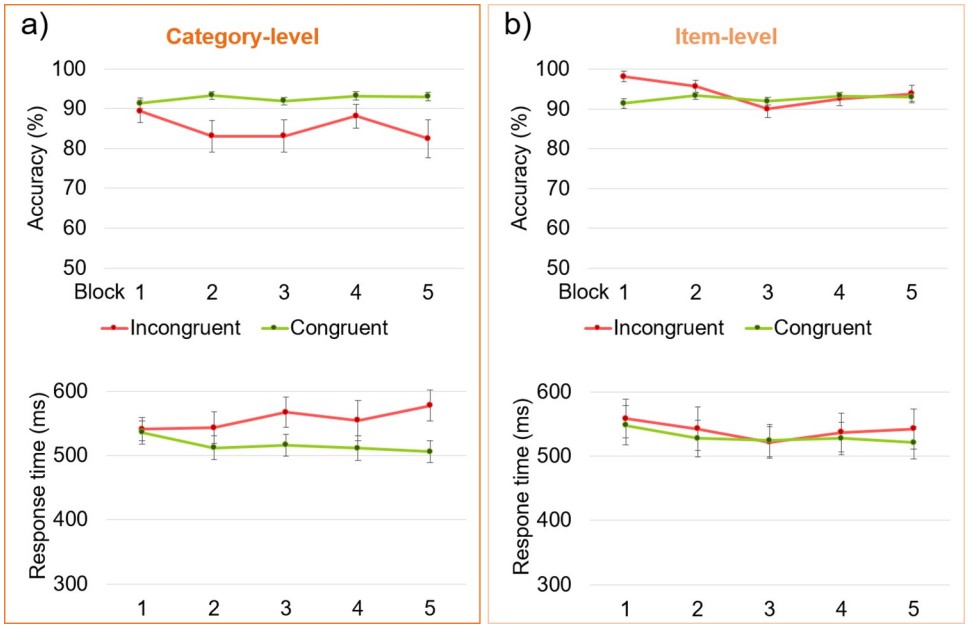

**Fig 4. Results across blocks (1–5).** Group means are plotted with error bars of ±1 SE. **a)** The category-level experiment, in which differences across incongruent and congruent trials increased after the first block. **b)** The item-level experiment.

learning, and the associations are more robust than in previous probability-based, category-level studies (single-exposure categorical words and colors: 2-11ms; 0.7–1.8%; [28]; non-word valence: 10-26ms; 0.5–2.9%; [29]; categorical word primes and target location: 5ms; [24]). Moreover, the current findings enable a novel, direct comparison of category-level with item-level associations: category-level effects were 31ms slower and 9.8% less accurate.

Importantly, stimulus-level and response-level learning cannot account for the category-level learning reported here. The sensory information (8 numbers paired with 2 colors) available was held constant across both experiment versions, suggesting that category-level associative learning occurred beyond the stimulus level. In suggestion that learning did not occur only at the response level (e.g., through motoric habituation, or through stimulus-response learning; in line with [22]), the explicit color association report accuracy was well-above chance level in the category-level experiment, despite participants using different response keys and their other hand. That is, even if participants did learn associations of numbers or categories with response keys and fingers, it is extremely unlikely that these associations would (systematically) transfer to different keys with a different spatial relationship, and fingers of a different hand. Similarly, although color corresponded with responses in the main experiment only in the category-level version (i.e., blue/down and yellow/up with a high probability), response-learning is unlikely to strengthen the dissociation between blue and yellow, or to become relevant for the task of determining a number's parity, so as to affect the results. Finally, the correlation of category-level participants' explicit color association report accuracy with their implicit difference scores, in terms of both accuracy and RT, indicates that participants who showed evidence of learning the associations between parity and color well in the main experiment were also better at reporting the high-probability color associations of individual numbers, despite changes in the response format.

Concept-level, associative learning has been supported in previous studies not using probability-based paradigms, e.g., in: 1) priming of a paired word-completion test [23]; 2) categorical

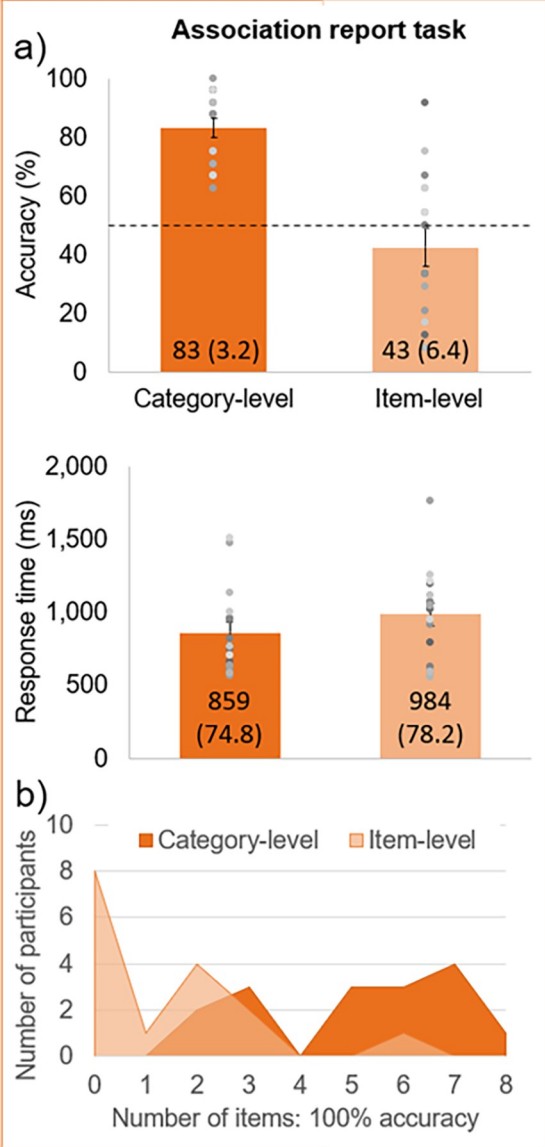

**Fig 5. Explicit, color association report task results. a)** In the category-level experiment, participants had a range of 62–100% accuracy; this range was 8–92% in the item-level experiment. Chance level at 50% is indicated with a dashed black line. Individual participants are represented as dots superimposed on the group-level mean bar plots, with error bars of ±1 SE (exact M and (SE) values are given on the bars). **b)** The number of participants achieving 100% accuracy per number of items (maximum 8) is plotted for both the category- and item-level experiments.

sequence order regularities [25–27]; and 3) visual search (within-category pictures speeding word responses: [51]; associated object-category color speeding object responses: [52]). In line with an automatic extraction of conceptual information, the Stroop effect occurs for color-associated words (e.g., lemon or sky; [53, 54]). It may be that perception has a fundamentally cognitive/conceptual nature (e.g., [31–34]), that is merely enhanced in synesthesia ([55–57]; it may be noted that high-probability color associations have been used to investigate the relation of implicit, associative learning to synesthesia [45, 46]).

One possibility is that concept-level (generalized) learning may have the same mechanisms as item-level (instance) learning, but the saliency of item-level or concept-level attributes

drives the level of learning ([58]; see also [59, 60]). Here, the concept of parity was salient: if participants were able to implicitly learn the association of parity with color, there are only two pairings necessary (blue/even and yellow/odd), instead of eight item-level pairings. Nonetheless, the co-occurrence of categorical stimuli may be a common phenomenon, and an important driver of associative learning, in the natural environment (e.g., [61]).

## Item-level associative learning

Associative learning was not clearly achieved in the item-level experiment: responses were 8.7ms slower and 1.5% *more* accurate to incongruent trials. Indeed, most participants (56%) who were slower were also more accurate; however, these small accuracy differences might be a ceiling effect (Fig 3). On the other hand, it is likely that implicit learning at the item level, which has been extensively demonstrated in previous studies, would have clearly occurred if more time was given, e.g., with more trials or sessions (e.g., [46]; see also [62, 63]). In particular, the 8 color-number pairings presented here are more than the 3–4 pairings typically used in previous studies reporting quickly onsetting effects (e.g., [10, 45]). It is possible that participants in the item-level experiment version did learn at least one color-number association, although not enough to drive learning effects overall (indeed, this appeared to be the case for half the participants in the explicit color association report task: **Fig 5B**). On the other hand, a relatively high proportion (~91%) of congruent trials was used here (see [64]), compared to previous studies (e.g., 75–83% congruent, with several incongruent colors: [11, 22]). Overall, the limitations of item-based learning here implies that views of predictive processing and statistical learning, which see associations as calculated predictors that facilitate behavioral performance ([15, 65]), should be extended beyond the item-level (for an attempt with simple artificial stimuli: [66]).

## Implicit learning and explicit report

Learning was apparent from the second block of trials in the category-level experiment (i.e., after the first block of ~100 congruent and 10 incongruent trials). This is consistent with previous studies, in which implicit learning is usually reported within or soon after the first block (i.e., after the first tens of trials; e.g., [11]; review: [12]; it should be noted that in some studies, the "first block" is designated after a number of practice trials). This rapidity is sometimes interpreted as an indicator of implicitness in learning [10]. Anecdotally, when participants were given the instructions for the color association report task, they often seemed concerned that they would not know how to respond; no participant spontaneously reported being aware of color-parity congruency, although they were not asked systematically.

Participants were yet able to give the associated color when explicitly asked in the category-level experiment (83% mean accuracy), although this cannot be taken as evidence that they were aware of the category-color association (see [30]). Indeed, if participants were explicitly aware that parity and color were associated, they were predicted to have near ceiling accuracy in this association report task. Only one participant reported the high-probability color associations with perfect accuracy, however; the average number of items with perfect accuracy was about 5 (out of 8; as compared to about 1 in the item-level experiment). Six items were reported with perfect accuracy by the best-performing participant in the item-level experiment (**Fig 5B**), underscoring that item-level associative learning likely did sometimes occur and that awareness of category-level relationships was not required for high explicit report performance.

## Color and number categories

Color and number may be a particularly good pair for establishing associations. Grapheme (including number)-color associations are a common form of synesthesia [67, 68], possibly

due to increased activations between nearby brain regions in the fusiform gyrus involved in color, number, and letter processing (e.g., [69, 70]). Indeed, despite some separate pathways for color and shape processing, these attributes are also bound (early) in human visual perception (e.g., [71, 72]). Color, in particular, has often been used as a paired attribute in previous studies, although this may also relate to paradigms' roots in the Stroop effect ([49]; reviews: [73, 74]).

Color and parity may also be a good pair for establishing category-level associations. In number-color synesthesia, synesthetic color associations appear to affect numerical cognition (magnitude: [75, 76]). Color is often learned and interacted with categorically (e.g., [77]; categorical effects in the word-reading Stroop task: [78, 79]). Here, blue and yellow were chosen as highly distinctive perceptual categories ([80]; neurally: [81]). Further, the color exemplars here were reasonably prototypical, which may facilitate their memory traces (in line with a prototypical bias of memory colors: [82, 83]; including that colors were not controlled for luminance or saturation, e.g., yellow was lighter than blue; [84, 85]). Parity is highly categorical (but see e.g. [86]), although other numerical concepts are more relative (e.g., magnitude). While numbers can be categorized in multiple ways, the parity task ensured that this concept was accessed here. Color and parity were not counter-balanced here, since a priori associations were not predicted: while there was no effect of congruency in the first category-level block, this could be further explored in future studies.

Finally, the finding that categorical consistency benefits associative learning could be extended in future studies to investigate whether the association of existing (color) concepts can facilitate learning of paired novel concepts, e.g., in mathematics and language learning. Color labeling has been applied to a wide range of topics, although an empirical basis and constraints for its validity has been lacking. Understanding the formation of conceptual relations, through further behavioral investigation as well as in terms of associative principles of the brain [87, 88], is an intriguing avenue for future research, with potential applications for improving learning strategies/materials.

## Supporting information

**S1 Fig. Implicit and explicit task correlations.** In the category-level experiment, individuals' explicit color association task accuracy (acc.) was significantly correlated with their difference scores in the main experiment, for both accuracy (congruent–incongruent) and RT (RT: incongruent–congruent).
(TIF)

## Acknowledgments

We thank Bruno Rossion and two anonymous reviewers for their feedback on an earlier version of this manuscript.

## Author Contributions

**Conceptualization:** Talia L. Retter, Lucas Eraßmy, Christine Schiltz.

**Data curation:** Talia L. Retter, Lucas Eraßmy.

**Formal analysis:** Talia L. Retter, Lucas Eraßmy.

**Funding acquisition:** Christine Schiltz.

**Investigation:** Talia L. Retter, Lucas Eraßmy.

**Methodology:** Talia L. Retter, Lucas Eraßmy, Christine Schiltz.

**Project administration:** Talia L. Retter, Christine Schiltz.

**Resources:** Christine Schiltz.

**Software:** Talia L. Retter.

**Supervision:** Talia L. Retter, Christine Schiltz.

**Visualization:** Talia L. Retter.

**Writing – original draft:** Talia L. Retter.

**Writing – review & editing:** Talia L. Retter, Lucas Eraßmy, Christine Schiltz.

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
