## [Decision Letter · Decision Letter 0]

17 May 2023

PONE-D-23-05869Categories facilitate implicit learning of color-number associationsPLOS ONE

Dear Dr. Retter,

Thank you for submitting your manuscript to PLOS ONE. After careful consideration, we feel that it has merit but does not fully meet PLOS ONE’s publication criteria as it currently stands. Therefore, we invite you to submit a revised version of the manuscript that addresses the points raised during the review process.

The reviewers thoroughly presented their comments. Please address the comments and improve the manuscript. 

We look forward to receiving your revised manuscript.

Kind regards,

Agnieszka Konys, Ph.D.

Academic Editor

PLOS ONE

“This work was supported by the Face Perception INTER project [INTER/FNRS/15/11015111 to CS], funded by the Luxembourgish Fund for Scientific Research (FNR, Luxembourg).”

“This work was supported by the Face Perception INTER project [INTER/FNRS/15/11015111 to CS], funded by the Luxembourgish Fund for Scientific Research (FNR, Luxembourg; http://www.fnr.lu/). The funders had no role in study design, data collection and analysis, decision to publish, or preparation of the manuscript.”

Reviewers' comments:

Reviewer's Responses to Questions

**Comments to the Author**

1. Is the manuscript technically sound, and do the data support the conclusions?

Reviewer #1: Yes

Reviewer #2: No

2. Has the statistical analysis been performed appropriately and rigorously? 

Reviewer #1: Yes

Reviewer #2: No

3. Have the authors made all data underlying the findings in their manuscript fully available?

Reviewer #1: Yes

Reviewer #2: No

4. Is the manuscript presented in an intelligible fashion and written in standard English?

Reviewer #1: Yes

Reviewer #2: Yes

5. Review Comments to the Author

Reviewer #1: The authors developed a simple but original paradigm to assess whether implicit learning of probabilistic associations between numbers and color categories can occur, and measure the magnitude of this putative learning effect. Their participants had to report the parity of numbers that were associated with two colors in a probabilistic way (e.g., even numbers more often with blue than yellow) or without any systematic association (control condition, different group of participants). They report a strong effect of systematic associations between colors and numbers, with participants being better and faster at judging parity for numbers that were congruently often associated with the same color. The effect emerges after a first block of trials and is stable throughout the experiment.

Despite color associations being explicitly retrieved (83% accuracy) in a post-experiment color association report.

The study is original and very well designed, with tight controls (also changing response keys and hands between the implicit learning association and the explicit color association report), and the results are clear, with important implications of theories of associative learning. I am not sure that opponents of conceptual influences of perception (e.g., Firestone & Scholl, 2016) would be convinced that these results support such a view, but the study deserves to be published regardless because it reports one of the strongest evidence of implicit learning of category-based associations, with practical implications for mathematics learning also. The paper is also very clearly written, with quality illustrations. I highly recommend it for publication and have only minor suggestions for revision.

- Title: replace ‘Categories’ with ‘Categorical consistency’ for clarity?

- Abstract; 5th sentence: what are low-probability trials (p=0.09)? This is not defined above and unclear, I was confused between high probability colors and high-probability trials, and it’s only when reading the methods that I understood.

I would suggest modifying the abstract as follow for clarity (also making it clear that the 2 experiments are in fact one experiment performed in two separate groups of participants):

In a category-level condition (participant group 1), even numbers (2,4,6,8) had a high-probability (p=.91) of appearing in blue, and odd numbers (3,5,7,9) in yellow. Associative learning was measured by the relative performance on low-probability (p=.09) to high-probability trials. There was strong evidence for associative learning: low-probability

32 performance was impaired (40ms RT increase and 8.3% accuracy decrease relative to high

probability). This was not the case in an item-level experiment (participant group 2) in which high-probability colors were non-categorically assigned (blue: 2,3,6,7; yellow: 4,5,8,9).

The categorical advantage was upheld in an explicit color association report (83% accuracy vs. 43% at the item-level). These results support …

- p.5. Conceptual nature of perception. Not sure Knowlton & Squire 1993 reference is appropriate there... Gregory instead? Gregory, R. (1970). The Intelligent Eye. London: Weidenfeld and Nicolson. Also Rock, I. (1993). The logic of Perception.

- p.8. What is ‘most often’? Please specify (even in introduction/abstract)

- p.9: parings  paIrings

- There were many more congruent trials than incongruent trials, and comparing performance to a random selection of congruent trials matching the number of congruent trials could have been more correct (is variance across the two types of trials not different?). However, this comparison was done in both conditions, so this cannot influence the results.

- Results, p.11. Remove: ‘In agreement with …’ to describe results here not interpretations

- In the item-level experiment, there were very small but significant differences between the two types of trials (in opposite direction for accuracy and RTs), which is surprising. Maybe these differences would not be found if the comparison was made between the same number of trials for each condition (see above)? This would make the results clearer. Combining accuracy scores and RTs in inverse efficiency scores (Townsend and Ashby (1978, 1983) could also help there, nullifying the effects (note that the difference for accuracy rates looks like a ceiling effect ...).

Reviewer #2: PONE-D-23-05869: Categories facilitate implicit learning of color-number associations

The paper reports a single two-group experiment (total N = 32) testing incidental learning of color-number associations. Participants were asked to report the parity of a digit on each trial (odd or even). One group had nearly all (91%) odd numbers in yellow and even numbers in blue. Consequently, there was 91% reliable mapping between color of the digit and the required response.

The results showed that participants in group 1 (named category-level in the paper) had decreased accuracy and increased reaction time when encountering exception items (9% of trials where odd numbers were blue or even numbers yellow). In other words, they experienced a conflict or interference between the color-typical response and the digit-required response. Participants noticed that the digits tended to be in a consistent color and had about 84% accuracy in reporting which color went with which digit in a subsequent explicit color-digit learning test. Furthermore, this explicit knowledge correlated with the accuracy and RT differences between the rule-following and exceptional color digits, suggesting that the more participants were aware of the digit-color rule, the more disruptive it was for them when the color was incongruent with the response (the digit parity).

Second group had nearly all (91%) digits 2,3,6,7 in blue and digits 4,5,8,9 in yellow. Consequently, the color of the digit and the required response were uncorrelated such that participants had to respond to half of the yellow digit with one response and the other half of the yellow digits with the other response.

In group 2 (named item-level condition in the paper), there wasn’t a clear association between color status (rule-following “congruent” or exceptional “incongruent”) on accuracy & RT of digit parity task. [BTW, To me, it did not make sense to even call the color-exception trials “incongruent” in this condition while there was clear incongruence (of a Stroop task type) in group 1 were the response required by the digit was clashing with the response typical for the color.] Participants in group 2 generally didn’t notice that some digits were consistently presented in one color while other digits in a different color (explicit color-digit task 43% correct, chance = 50%). Finally, there was no correlation between (non-existent) interference effect in the parity task and the (non-existent) explicit learning of color-digit associations.

I was excited to read this article based on the title, but unfortunately it did not meet my expectations. While my review is ultimately quite negative, I hope it provides feedback that will be useful in the authors future endevors.

Major major comment.

The article is framed around the benefit of category consistency but I don’t think this design allows us to make such conclusions because the category-level vs. item-level manipulation is 100% confounded with task-relevance. Color was a helpful clue in category-level condition, so it makes sense that participants picked up on it. It was useless (task irrelevant) in item-level condition, so it makes sense that participants would try to ignore it. There is a lot of literature on how memory for task-irrelevant info is poor and for task-relevant info is better, even in incidental learning paradigms. It’s a trivial finding as it stands. It would have been a lot more interesting if the conditions were equated with respect to task relevance (e.g., task is odd/even and colors are small/large or task is small/large and colors are odd/even). That way, the color would irrelevant to the task at hand in both conditions, but one may incidentally pick up on the regularity when there is a rule to the regularity (category consistency) compared to when it is item-unique. THEN it would be possible to make stronger claims about the benefits of category consistency over item-level assignment. Not so with the current task as it all can be driven by task relevance vs. irrelevance.

Other major comments.

The issue above, by itself, is in my view a key flaw of this design to answer the stated question. But for future work, I wanted to point out some of the other issues.

1. The design was a factorial 2x2 design with color-digit grouping (category-level vs. item-level) as a between subjects factor and trial type (congruent vs. incongruent) as a within-subject factor. It would be helpful if all figures also have 2x2 format so one can better compare across conditions/groups. In this case, the lack of comparison across groups makes it harder to see the full pattern of results that may not really align with the concluding claims, even if the task itself wasn’t flawed. The within-group comparisons did not directly support the conclusions terms of benefits of scaffolding new learning by existing categories. In fact, the results mainly showed how disruptive it was in category-level (group 1) condition when the digit response and the typical color response were incongruent. For instance, Figure 4a (accuracy plots) shows essentially identical accuracies in all conditions EXCEPT category-level condition incongruent trials where accuracy is impaired. Thus, the actual pattern of results here does NOT support the conclusion claims regarding category-related benefits. Having the data in a single 2x2 plot and analyzing them with the intended conclusions in mind would make it more explicit. [BTW, There is research in educational psychology showing that adding extraneous information to graphs that maybe helps with a given graph can IMPAIR general graph reading. This seems to be the same story.]

2. There were other unsupported conclusions (conceptual nature of perception, color categories facilitating learning of novel concepts, etc). Not that those claims are in themselves incorrect, but this study does not speak to them.

3. The sample size is too small for a simple cognitive experiment with healthy young adults. Having prior poorly powered studies does not mean that future studies can all have equally small N.

4. Many statistical analyses are not appropriate. Despite the tiny sample size, the authors analyze everything with parametric tests inappropriate for such small N and skewed data (anovas, t-tests, pearson’s r). They repeatedly list absolute effect sizes (40ms, 8.3% accuracy decrement) without providing confidence intervals for them, which would make them much less impressive (see the skew and individual differences in Fig2a, showing the need to use some sort of robust non-parametric statistical approaches). They don’t describe what error bars on graphs represent, but they certainly do not represent across-subject standard error of the mean (maybe they are within-subject SEMs??). Cohen’s D is not an appropriate measure of effect size for a paired t-test.

5. The authors often use the word “implicit” unnecessarily and sometimes incorrectly. Learning was “incidental”, not “implicit” (as subjects had explicit knowledge of color-digit associations), and it is not appropriate to just say that by “implicit” they mean “incidental”. One term is clear and well defined (“incidental” learning without explicit instruction), the other is a lot more convoluted and highly debated (as the authors themselves acknowledge), and not fitting at all the current data. One can measure learning implicitly (as was done with RT & accuracy effects in the parity task), but that does not mean that the knowledge itself is implicit. In fact, the positive correlation between the implicit measure and the explicit knowledge rules out that the learning was implicit, in the sense how that term is typically used.

6. The authors should use median RTs as participant’s RT measures, if they haven’t done that. This alleviates any need for excluding any trials based on long RTs. Excluding correct RTs that were too long (lines 197-198) may be biasing the RT results when there are accuracy differences between conditions.

Small comment:

1. The authors report they will not share the data freely, but no justification is provided. Again, for a simple cognitive task with no sensitive questions, that is difficult to justify.

2. It does not make sense to require 100% performance on the explicit color-digit test as a proof of explicit knowledge. Participants didn’t even have 100% performance on digit parity task, while they presumably have a perfect knowledge of which digit is odd and which even.

3. The authors mean something very narrow and specific when they say “probability-based learning” but don’t explain it, so e.g. lines 70-76 are not readily comprehensible. There are several other places where writing could be more accessible to a reader not familiar with prior studies with similar paradigms. BTW, Many others study probabilistic/statistical learning in different ways, and it would be good to consider that work here as well.

4. The authors are linking their work to synesthesia, but again this is not well explained. BTW, There are interesting studies on synesthesia how individual cases match a color-letter associations from childhood toys, indicating that at least some synesthesia cases are driven by learning and long-term exposure to consistent color-letter associations. The authors could consider those findings in their future research.

5. Why were instructions in English when participants were not English speakers?

6. The task description (especially timing, trial counts, etc) could improve in clarity and readability.

7. The 91%/9% ratio seems unnecessary extreme, leaving few rule-breaking trials for analysis. The low within-subject power is then combined with low across-subjects power. Under these conditions, one subject’s data (such as the extreme subject with 30% accuracy drop and 200ms RT cost, Figure 3) may have unruly influence on the overall results.

8. Figure 5b would be easier read in a standard histogram format.

6. PLOS authors have the option to publish the peer review history of their article (what does this mean?). If published, this will include your full peer review and any attached files.

Reviewer #1: No

Reviewer #2: No

---

## [Author Response · Author response to Decision Letter 0]

14 Jun 2023

Response to Reviewers : PONE-D-23-05869

Reviewer #1: The authors developed a simple but original paradigm to assess whether implicit learning of probabilistic associations between numbers and color categories can occur, and measure the magnitude of this putative learning effect. Their participants had to report the parity of numbers that were associated with two colors in a probabilistic way (e.g., even numbers more often with blue than yellow) or without any systematic association (control condition, different group of participants). They report a strong effect of systematic associations between colors and numbers, with participants being better and faster at judging parity for numbers that were congruently often associated with the same color. The effect emerges after a first block of trials and is stable throughout the experiment. Despite color associations being explicitly retrieved (83% accuracy) in a post-experiment color association report.

The study is original and very well designed, with tight controls (also changing response keys and hands between the implicit learning association and the explicit color association report), and the results are clear, with important implications of theories of associative learning. I am not sure that opponents of conceptual influences of perception (e.g., Firestone & Scholl, 2016) would be convinced that these results support such a view, but the study deserves to be published regardless because it reports one of the strongest evidence of implicit learning of category-based associations, with practical implications for mathematics learning also. The paper is also very clearly written, with quality illustrations. I highly recommend it for publication and have only minor suggestions for revision.

Reply: We thank Reviewer #1 for the positive evaluation of our manuscript. Please find our replies below, with indicated page numbers referring to the manuscript version with tracked changes.

- Title: replace ‘Categories’ with ‘Categorical consistency’ for clarity?

Reply: We have modified the title as suggested.

- Abstract; 5th sentence: what are low-probability trials (p=0.09)? This is not defined above and unclear, I was confused between high probability colors and high-probability trials, and it’s only when reading the method/s that I understood.

Reply: Thank you, we have changed this description to: “…relative performance on trials with low-probability (p=.09) to high-probability (p=.91) number colors”.

I would suggest modifying the abstract as follow for clarity (also making it clear that the 2 experiments are in fact one experiment performed in two separate groups of participants):

In a category-level condition (participant group 1), even numbers (2,4,6,8) had a high-probability (p=.91) of appearing in blue, and odd numbers (3,5,7,9) in yellow. Associative learning was measured by the relative performance on low-probability (p=.09) to high-probability trials. There was strong evidence for associative learning: low-probability performance was impaired (40ms RT increase and 8.3% accuracy decrease relative to high probability). This was not the case in an item-level experiment (participant group 2) in which high-probability colors were non-categorically assigned (blue: 2,3,6,7; yellow: 4,5,8,9). The categorical advantage was upheld in an explicit color association report (83% accuracy vs. 43% at the item-level). These results support …

Reply: We agree that it is more clear to separate the two experiments in the abstract, and have followed that suggestion. We also specify that the item-level experiment was performed by a different group of participants. 

- p.5. Conceptual nature of perception. Not sure Knowlton & Squire 1993 reference is appropriate there... Gregory instead? Gregory, R. (1970). The Intelligent Eye. London: Weidenfeld and Nicolson. Also Rock, I. (1993). The logic of Perception.

Reply: We have changed the references as suggested. We were familiar with Gregory’s writing but had not yet read Rock’s, thank you.

- p.8. What is ‘most often’? Please specify (even in introduction/abstract)

Reply: We have removed the description of “most often”, and now simply refer to high-probability and low-probability trials throughout the manuscript (e.g., p. 8).

- p.9: parings  paIrings

Reply: This typo has been fixed, thanks.

- There were many more congruent trials than incongruent trials, and comparing performance to a random selection of congruent trials matching the number of congruent trials could have been more correct (is variance across the two types of trials not different?). However, this comparison was done in both conditions, so this cannot influence the results.

Reply: We acknowledge that the number of trials is inherently different for high-probability congruent than low-probability incongruent trials. While we could match the number of trials by selecting a subset of congruent trials only, this might incidentally bias the results. We agree that in this regard the analysis relies on comparing the experimental and control experiment versions.

- Results, p.11. Remove: ‘In agreement with …’ to describe results here not interpretations

Reply: This has been removed.

- In the item-level experiment, there were very small but significant differences between the two types of trials (in opposite direction for accuracy and RTs), which is surprising. Maybe these differences would not be found if the comparison was made between the same number of trials for each condition (see above)? This would make the results clearer. Combining accuracy scores and RTs in inverse efficiency scores (Townsend and Ashby (1978, 1983) could also help there, nullifying the effects (note that the difference for accuracy rates looks like a ceiling effect ...).

Reply: We consider that small differences in the item-level experiment are an actual part of the data that is worth reporting. We are aware of inverse efficiency scores as a useful measure for combining accuracy and RT, but think that it makes sense to report both measures separately here, in the absence of an apparent speed-accuracy tradeoff at either experiment level (we agree that the item-level accuracy difference is likely a ceiling effect, being of only 1.46%). This is mentioned in the revised discussion (p. 19).

We thank Reviewer # 1 for the constructive comments which have helped improved the manuscript.

Reviewer #2 : 

The paper reports a single two-group experiment (total N = 32) testing incidental learning of color-number associations. Participants were asked to report the parity of a digit on each trial (odd or even). One group had nearly all (91%) odd numbers in yellow and even numbers in blue. Consequently, there was 91% reliable mapping between color of the digit and the required response.

The results showed that participants in group 1 (named category-level in the paper) had decreased accuracy and increased reaction time when encountering exception items (9% of trials where odd numbers were blue or even numbers yellow). In other words, they experienced a conflict or interference between the color-typical response and the digit-required response. Participants noticed that the digits tended to be in a consistent color and had about 84% accuracy in reporting which color went with which digit in a subsequent explicit color-digit learning test. Furthermore, this explicit knowledge correlated with the accuracy and RT differences between the rule-following and exceptional color digits, suggesting that the more participants were aware of the digit-color rule, the more disruptive it was for them when the color was incongruent with the response (the digit parity).

Second group had nearly all (91%) digits 2,3,6,7 in blue and digits 4,5,8,9 in yellow. Consequently, the color of the digit and the required response were uncorrelated such that participants had to respond to half of the yellow digit with one response and the other half of the yellow digits with the other response.

In group 2 (named item-level condition in the paper), there wasn’t a clear association between color status (rule-following “congruent” or exceptional “incongruent”) on accuracy & RT of digit parity task. [BTW, To me, it did not make sense to even call the color-exception trials “incongruent” in this condition while there was clear incongruence (of a Stroop task type) in group 1 were the response required by the digit was clashing with the response typical for the color.] Participants in group 2 generally didn’t notice that some digits were consistently presented in one color while other digits in a different color (explicit color-digit task 43% correct, chance = 50%). Finally, there was no correlation between (non-existent) interference effect in the parity task and the (non-existent) explicit learning of color-digit associations.

I was excited to read this article based on the title, but unfortunately it did not meet my expectations. While my review is ultimately quite negative, I hope it provides feedback that will be useful in the authors future endevors.

Reply: We hope to positively surprise Reviewer #2 by our ability to respond to the review in the context of the present endeavor. We appreciate the criticisms which have led to the improvement of the manuscript, and thank Reviewer #2 sincerely. Please find our replies below, with the indicated page numbers referring to the version of the manuscript with tracked changes.

Major major comment.

The article is framed around the benefit of category consistency but I don’t think this design allows us to make such conclusions because the category-level vs. item-level manipulation is 100% confounded with task-relevance. Color was a helpful clue in category-level condition, so it makes sense that participants picked up on it. It was useless (task irrelevant) in item-level condition, so it makes sense that participants would try to ignore it. There is a lot of literature on how memory for task-irrelevant info is poor and for task-relevant info is better, even in incidental learning paradigms. It’s a trivial finding as it stands. It would have been a lot more interesting if the conditions were equated with respect to task relevance (e.g., task is odd/even and colors are small/large or task is small/large and colors are odd/even). That way, the color would irrelevant to the task at hand in both conditions, but one may incidentally pick up on the regularity when there is a rule to the regularity (category consistency) compared to when it is item-unique. THEN it would be possible to make stronger claims about the benefits of category consistency over item-level assignment. Not so with the current task as it all can be driven by task relevance vs. irrelevance.

Reply: We acknowledge that in our design color correlated with the response keys with a high-probability in the category-level and not item-level experiment. However, we do not agree that this makes color task-relevant. Importantly, the task is to report the parity of the numbers, not to learn a novel, arbitrary category of numbers. If we had asked participants to perform a task on identifying a “category” of 2,3,6,7, as was associated with blue in the item-level experiment, then we can understand Reviewer #2’s concern that participants may require effortful memory to perform the task, and then may become consciously aware of the relationship with blue and use color in task performance. However, we do not think this was the case with the parity task as used here: participants did not have to remember experimental information to perform the task, they simply had to access long-term numerical knowledge of parity that is rapidly, and arguably automatically, accessed. Importantly, the participants in both experiment level groups performed the parity task rapidly and with high accuracy (93-94% accuracy and 515-530 ms RTs for congruent trials across both experiment levels): we do not find any evidence in the data that in addition to evaluating the parity of the number they also evaluated its color in the category-level experiment only. Moreover, we think that participants did not have an awareness of the color association of the numbers (please see our reply to point 5 below). We address this issue in the Discussion of the revised manuscript (p. 18).

While we agree that using an orthogonal task, like a small/large task, would ensure that color did not correlate with the response key in the category-level experiment, it would likely not enable the assessment of implicit learning of color-number associations, as it would make the targeted Stoop-like incongruency conflict between colors and numbers irrelevant. In analogy, having a small/large task in the classic Stroop design of colored color words (or implicitly learned color-word associations), instead of a related color task, would likely not enable the targeted conflict measurement, and is not the experimental standard. 

Other major comments.

The issue above, by itself, is in my view a key flaw of this design to answer the stated question. But for future work, I wanted to point out some of the other issues.

1. The design was a factorial 2x2 design with color-digit grouping (category-level vs. item-level) as a between subjects factor and trial type (congruent vs. incongruent) as a within-subject factor. It would be helpful if all figures also have 2x2 format so one can better compare across conditions/groups. In this case, the lack of comparison across groups makes it harder to see the full pattern of results that may not really align with the concluding claims, even if the task itself wasn’t flawed. The within-group comparisons did not directly support the conclusions terms of benefits of scaffolding new learning by existing categories. In fact, the results mainly showed how disruptive it was in category-level (group 1) condition when the digit response and the typical color response were incongruent. For instance, Figure 4a (accuracy plots) shows essentially identical accuracies in all conditions EXCEPT category-level condition incongruent trials where accuracy is impaired. Thus, the actual pattern of results here does NOT support the conclusion claims regarding category-related benefits. Having the data in a single 2x2 plot and analyzing them with the intended conclusions in mind would make it more explicit. [BTW, There is research in educational psychology showing that adding extraneous information to graphs that maybe helps with a given graph can IMPAIR general graph reading. This seems to be the same story.]

Reply: The figures fully report the data, plotting incongruent and congruent trials for both the category-level and item-level experiments (separately in Figs. 2 and 4, but this nevertheless provides the equivalent information), including individual participants’ data points, with no extraneous information (it may be noted that the illustrations were praised by Reviewer #1). It is indeed apparent from the figures (and reported data) that incongruent performance is worse than congruent performance for the category-level experiment: that is, that categorical consistency facilitated associative learning, as evidenced by a Stroop-like interference effect for incongruent vs. congruent trials. We do not conclude or ever mention “category-related benefits”. Instead, in the abstract we write transparently that “low-probability performance was impaired”. This is in line with the presence of generally stronger interference effects than facilitation effects in color-related Stroop-like designs (e.g., Hershman, Beckmann, & Henik, 2022, Psychophysiology). Data combining both experiment levels are plotted in Figs. 3 and 5b.

2. There were other unsupported conclusions (conceptual nature of perception, color categories facilitating learning of novel concepts, etc). Not that those claims are in themselves incorrect, but this study does not speak to them.

Reply: The topics mentioned above were not given as conclusions in the manuscript, but as relevant topics. Whether perception has a conceptual nature or not is an important part of the introductory framework for this study, and was largely addressed in the Introduction of this manuscript. We think that a categorical facilitation of implicit learning, extending beyond the individual item level, would indeed be in line with a theoretical framework of a conceptual nature of perception, although our study certainly does not provide conclusive evidence for that.

In regards to novel concepts, we merely hypothesize in two sentences that : “More practically, it might imply that a well-known concept (color categories) might be applied to facilitate learning of a novel concept (e.g., parity categories for young children)” (p. 5); and that “Finally, the finding that categorical consistency benefits associative learning could be extended in future studies to investigate whether the association of existing (color) concepts can facilitate learning of paired novel concepts, e.g., in mathematics and language learning.” (p. 22). We were careful to present such topics as tentative hypotheses, and not conclusions, throughout the revised manuscript.

3. The sample size is too small for a simple cognitive experiment with healthy young adults. Having prior poorly powered studies does not mean that future studies can all have equally small N.

Reply: The same size was justified in relation to previous studies similarly measuring implicit associative learning that reported robust effect sizes (p. 6). 

4. Many statistical analyses are not appropriate. Despite the tiny sample size, the authors analyze everything with parametric tests inappropriate for such small N and skewed data (anovas, t-tests, pearson’s r). They repeatedly list absolute effect sizes (40ms, 8.3% accuracy decrement) without providing confidence intervals for them, which would make them much less impressive (see the skew and individual differences in Fig2a, showing the need to use some sort of robust non-parametric statistical approaches). They don’t describe what error bars on graphs represent, but they certainly do not represent across-subject standard error of the mean (maybe they are within-subject SEMs??). Cohen’s D is not an appropriate measure of effect size for a paired t-test.

Reply: We think that parametric tests are appropriate for our data, and note that Reviewer # 2 does not propose any alternative analyses. We tested for equality of variance across samples using Leven’s test and corrected the degrees of freedom accordingly (p. 11). We comprehensively reported the magnitude of effects in our data (e.g., 40ms) with standard errors, and also the statistics including effect sizes (e.g., Cohen’s d, about which we disagree with Reviewer # 2, and think that it is in fact an appropriate measure of effect size for a paired t-test, there being a specific formula for this, as standardly applied in SPSS as used here). We describe the error bars on every figure: e.g., for Fig. 2: “…group-level mean bar plots, with error bars of ±1 SE...”. The error bars do indeed represent across-subject standard error of the mean: perhaps they appear small as the range on the y-axis is larger than what Reviewer # 2 is used to (e.g., ranging from 50-100% accuracy).

5. The authors often use the word “implicit” unnecessarily and sometimes incorrectly. Learning was “incidental”, not “implicit” (as subjects had explicit knowledge of color-digit associations), and it is not appropriate to just say that by “implicit” they mean “incidental”. One term is clear and well defined (“incidental” learning without explicit instruction), the other is a lot more convoluted and highly debated (as the authors themselves acknowledge), and not fitting at all the current data. One can measure learning implicitly (as was done with RT & accuracy effects in the parity task), but that does not mean that the knowledge itself is implicit. In fact, the positive correlation between the implicit measure and the explicit knowledge rules out that the learning was implicit, in the sense how that term is typically used.

Reply: We are aware of a disagreement among researchers about the terms “implicit” and “incidental”. Unfortunately we are not on the same side of this debate as Reviewer # 2. However, we give our definition of “implicit” explicitly in the first paragraph of the Introduction: “…here we define “implicit” associative learning simply as learning that occurs incidentally, without explicit instruction or evidence of awareness during learning.” (p. 3). We do not agree that participants had explicit knowledge of color-number associations, despite the correlation of their accuracy and RT effects with the explicit color report task in the category-level experiment version, as extensively discussed in the manuscript (particularly in the Discussion section “Implicit learning and explicit report”, p. 20; please also see the reference Jiminez, Mendez & Cleermans, 1996). Participants may be able to correctly report the associated color of a number without having been aware of the color association, and many participants did not have high explicit color report accuracy for most individual number items. 

6. The authors should use median RTs as participant’s RT measures, if they haven’t done that. This alleviates any need for excluding any trials based on long RTs. Excluding correct RTs that were too long (lines 197-198) may be biasing the RT results when there are accuracy differences between conditions.

Reply: We think that mean RTs are an appropriate measure of participants’ response time, and that excluding long RT trials (>2.5 SDs from the individual participant’s mean) is less likely to bias the results than retaining them. We acknowledge the concern that excluding trials may affect the data, but have calculated that this affected only 2.3% of trials on average, and have added this information to the revised manuscript (p. 10).

Small comment:

1. The authors report they will not share the data freely, but no justification is provided. Again, for a simple cognitive task with no sensitive questions, that is difficult to justify.

Reply: We indicated that the data were not available without restrictions at the time of submission, but the data will be shared upon publication: we state in the cover letter of our resubmission that “The data will be available upon publication at the following repository address: https://zenodo.org/deposit/5913254.”

2. It does not make sense to require 100% performance on the explicit color-digit test as a proof of explicit knowledge. Participants didn’t even have 100% performance on digit parity task, while they presumably have a perfect knowledge of which digit is odd and which even.

Reply: We agree, we do not require 100% performance for proof of explicit knowledge. Our point was that even if participants’ accuracy appears high overall, it is not near-ceiling, and that most participants do not have 3/3 correct responses for many individual stimuli (p. 5; p. 21), suggesting imperfect learning; on the other hand, an item-level stimulus analysis allowed us to probe performance more closely in both experiment levels, also revealing that accuracy was sometimes high for all stimuli even in the item-level experiment.

3. The authors mean something very narrow and specific when they say “probability-based learning” but don’t explain it, so e.g. lines 70-76 are not readily comprehensible. There are several other places where writing could be more accessible to a reader not familiar with prior studies with similar paradigms. BTW, Many others study probabilistic/statistical learning in different ways, and it would be good to consider that work here as well.

Reply: Probability-based learning is defined earlier, on lines 55-57. We reference a number of different methods for studying implicit associative learning in the revised manuscript (p. 4): “…priming: Schacter & Graf, 1986; Lambert & Sumich, 1996; sequence order regularities: Hartman, Knopman & Nissen, 1989; Goschke & Bolt, 2007; Brady & Oliva, 2008; word categories: Schmidt et al., 2018)”. The topic of other designs in statistical learning is also returned to in the Discussion (p. 18).

4. The authors are linking their work to synesthesia, but again this is not well explained. BTW, There are interesting studies on synesthesia how individual cases match a color-letter associations from childhood toys, indicating that at least some synesthesia cases are driven by learning and long-term exposure to consistent color-letter associations. The authors could consider those findings in their future research.

Reply: While we do not link our work to synesthesia, we mention that some studies have compared learned, implicit color associations from an experiment setting with the color associations of synesthetic participants (e.g., Kusnir & Thut, 2012; Bankieris & Aslin, 2017; p. 19). We are aware of a famous study on synesthetic perceptions matching childhood toy colors (Witthoft & Winawer, 2013, Psychological Science), as well as studies contradicting such an apparent simplicity (e.g., Rich, Bradshaw & Mattingley, 2005, Cognition; Weiss, Greenlee & Volberg, 2018, i-Perception), and agree that it is for future research to continue that particular debate.

5. Why were instructions in English when participants were not English speakers?

Reply: The participants all understood English, although they also spoke other languages as the study took place in a multi-lingual country. We now specify that “Written experiment information and instruction was in English, which was spoken by all participants…” (p. 7).

6. The task description (especially timing, trial counts, etc) could improve in clarity and readability.

Reply: We are not aware of what is not clear about the task description in the text. However, the stimulation parameters and task are also illustrated in Figure 1.

7. The 91%/9% ratio seems unnecessary extreme, leaving few rule-breaking trials for analysis. The low within-subject power is then combined with low across-subjects power. Under these conditions, one subject’s data (such as the extreme subject with 30% accuracy drop and 200ms RT cost, Figure 3) may have unruly influence on the overall results.

Reply: The difference scores referred to in Figure 3 reflect both incongruent and congruent trial performance, such that the number of incongruent trials does not explain the variability across participants. Further, in the item-level experiment there was little variability across participants, despite the same number of incongruent trials as in the category-level experiment. While a 10:1 ratio as used here is relatively high compared to previous studies, we wanted favorable conditions for participants learning the high-probability associations, given the relatively higher number of stimulus associations to be learned (discussed with references on p. 19-20).

8. Figure 5b would be easier read in a standard histogram format.

Reply: We chose overlapping histogram plots across category-level and item-level groups so that these group distributions can be easily compared.

We thank Reviewer # 2 again for the critical comments that have led to the amelioration of the manuscript.

---

## [Editor Report · Decision Letter 1]

22 Jun 2023

Categorical consistency facilitates implicit learning of color-number associations

PONE-D-23-05869R1

Dear Dr. Retter,

We’re pleased to inform you that your manuscript has been judged scientifically suitable for publication and will be formally accepted for publication once it meets all outstanding technical requirements.

Kind regards,

Agnieszka Konys, Ph.D.

Academic Editor

PLOS ONE
---

## [Editor Report · Acceptance letter]

30 Jun 2023

PONE-D-23-05869R1 

Categorical consistency facilitates implicit learning of color-number associations 

Dear Dr. Retter:

I'm pleased to inform you that your manuscript has been deemed suitable for publication in PLOS ONE. Congratulations! Your manuscript is now with our production department. 

Kind regards, 

on behalf of

Dr. Agnieszka Konys 

Academic Editor

PLOS ONE